# The Potential of Digitally Enabled Disaster Education for Sustainable Development Goals

Mihoko Sakurai [1],* and Rajib Shaw [2]

1    Center for Global Communications, International University of Japan, Minato-ku, Tokyo 106-0032, Japan
2    Graduate School of Media and Governance, Keio University, Fujisawa 252-0882, Japan; shaw@sfc.keio.ac.jp
*    Correspondence: msakurai@glocom.ac.jp

**Abstract:** A sustainable and resilient local community requires a learning culture that allows them to evolve over time. Disaster education in this context is expected to be an important element for local communities. Conventionally, disaster education in Japan is provided in elementary and junior high school as an evacuation drill. After that age, the attachment with the local community becomes relatively low, which we call the black box of disaster education. This paper reports on a practical research project in Muroran City, Japan. It aimed to use digital technology to involve high school students in a disaster education program. Officials in Muroran City have been struggling with collecting young people to participate in a community leader development program for disaster risk reduction (DRR). The research project employed a cloud-based learning platform in order to appeal to high school students. A set of three workshops was conducted from November to December 2021. Three out of the five categories of DRR consciousness increased after the workshop, namely, imagination, mutual aid and interest. We observed that participants' mindsets and behaviors changed during the workshop activities. Digital technology can contribute to context-specific disaster risk education, which we believe is important in designing a sustainable and resilient local community for the 2030s.

**Keywords:** disaster risk reduction; education; digital technology; local community; sustainability

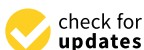

## 1. Introduction

In the dynamic global risk landscape, it is important to look at the complexity of the interplay of different types of risk in society. While we started 2015 with several global frameworks, such as the Sustainable Development Goals (SDGs), the Paris Agreement on Climate Change, the Sendai Framework for Disaster Risk Reduction (SFDRR), the Urban Agenda, etc., the global pandemic from 2020 onward has shaken many of these development goals. The World Economic Forum (WEF) Global Risk Outlook [1] has shown that environmental risk dominates in terms of both likelihood and impacts in the global risk landscape. Due to the pandemic, infectious diseases, which were considered earlier as a risk with low impact, mostly occurring in developing or under-developed countries, has now taken the top place in terms of impact. New risks such as digital power concentration and digital inequality are getting global attention in terms of their future likelihood. The same report from 2022 [2] showed that, in spite of the global pandemic taking the majority of attention in the last two years, climate- or environment-related hazards still top the risk landscape, both in terms of likelihood and impact. This was also stated and exemplified in the recent IPCC report from Working Group II [3], which talks about the urgency of action for climate-related hazards, including different types of climate disasters such as typhoons, floods, landslides, heat waves, etc. All of these combinations of different types of risks, both existing and emerging, have produced a critical challenge for the people and communities living in different ecosystems, be it in urban, rural, coastal, mountain or arid areas.

The importance of human behavior, its sensitivity to informed decision making and action, has become more relevant than ever. Knowledge, perception, behavior and actions

are critical to this. While traditional knowledge and community behavior is essential, the development of new technologies is providing us with new sets of information and solutions in the new digital era [4]. The youth of today are the leaders of tomorrow; so-called "Generation Z", who were born between the mid-1990s and 2010, are considered to be digital natives [5]. This is the first social generation that has grown up with access to the internet and portable digital technology. Thus, digital technology plays a critical role in all aspects of sociocultural and development perspectives. The governance mechanisms to support the technology are still developing at a slower pace than the technology growth [6]. However, we cannot deny the roles played by different types of emerging and new technologies in this digital era [7] and the role of governance mechanisms in supporting innovative resilience activities in local communities [8,9]. Keeping this in mind, this paper introduces practical research on digitally enabled disaster education for SDG 11: make cities and human settlements inclusive, safe, resilient and sustainable.

Japan is one of the most disaster-prone countries in the world. Located on the plate boundaries, the country is prone to earthquakes, tsunamis and volcanic eruptions. Japan is also at high risk from typhoons and flood disasters, which are annual events, and the recent climate change report by the IPCC emphasizes higher intensity and frequency in future [3]. In this paper, digital-based education for disaster risk reduction (DRR) is discussed. In 2017, the United Nations Office for Disaster Risk Reduction (UNDRR) defined disaster risk reduction as "*preventing new and reducing existing disaster risk and managing residual risk, all of which contribute to strengthening resilience and therefore to the achievement of sustainable development*" [10]. Seventy percent of Japan's land area is covered by mountains, which cause landslide risks. While the country is prone to different types of hazards, it has also built its capacity over the years through governance, innovation in technology and education and awareness of people and communities. "Hazard" in this context is defined by the UNDRR as "*a process, phenomenon or human activity that may cause loss of life, injury or other health impacts, property damage, social and economic disruption or environmental degradation*". Several previous studies have emphasized that the role of disaster education has become critical for every aspect of life in Japan [11–14]. The roles of schools and communities have become critical for effective disaster education. While disaster education is taught in different types of curricular and extra-curricular activities in primary and secondary education, there is a gap of several years in the Japanese system. We call this gap the black box of disaster education, which can spread for 10–15 years depending on the context [13]. Usually, the students undergo some sort of disaster education in elementary and junior high school until the age of 15 years. However, when it comes to high school and above, the attachment with the community becomes relatively low. After university graduation, when a person starts their professional career, the same detachment from the community exists.

In the Great Hanshin Awaji Earthquake of 1995, there were two age groups that were affected severely. One was the old generation, more than 65 years old (due to the inherent nature of the buildings and social vulnerability), and the other was the younger age group in their 20s and 30s. People in the younger age group lived alone in their apartments and had little interaction with the community [12]. This younger generation becomes attached to the community after marriage and having children, especially through different types of activities with kindergartens. Therefore, usually, 15 to 30 age group is detached from the community; however, they are critical in contributing to different types of community activities in the disaster education context. Although this was observed more than 25 years ago, it was also reflected in the recent disasters such as the 2011 East Japan Earthquake and Tsunami, where the younger generation was not aware of the evacuation places compared to the older generation. This was also reflected in the recent typhoon disaster in 2020 in the Kanto region, where significant parts of the urban population were affected [11].

In community activities, typically, it is the aged population that is active, be it the resident association (*Jichikai* in Japanese), voluntary disaster prevention organization (*Jishubo* in Japanese) or any other type of voluntary group, such as the fire volunteers (*Shobodan* in Japanese), flood volunteers (*Suibodan* in Japanese), etc. It is always a challenge to develop

inter-generational collective actions bringing different age groups together for community activities, especially those targeting DRR and development in resilient communities. This study aimed to use digital technology to involve high school students in disaster risk reduction-related community activities. Through this attempt, we tried to understand what digital technology can do for building-disaster-resilient communities. We define resilience as the "*adaptive capacity of an organization in a complex and changing environment*" [15]. We believe a resilient community should obtain such a capacity for future disasters.

First, we introduce the theoretical background of this research, discussing the state of the art of DRR education in Japan and how digital technology relates to the implementation of SDGs. The research design is presented, followed by the research context and instruments. This paper presents a practical DRR education case, specifically a series of workshops with digital technology in a municipality in Japan. An analysis part shows both the qualitative and quantitative results of the workshop. The last part of this paper presents the discussion and conclusion. Both practical and theoretical implications are discussed. The case shows that a digital DRR education program succeeded in getting high school students involved in DRR activities and the workshop changed their awareness towards DRR. From an international perspective, as a disaster-prone country, Japan needs to respond to issues reported by the WEF and IPCC. A practical case study of Japanese digital DRR education activities could enhance understanding of the use of digital technologies for future preparedness in other disaster-prone countries. From a national perspective, on the other hand, the results of the workshop provide us with a clue for how to approach the black box gap of disaster education by utilizing digital technologies. We believe this paper could cultivate possibilities for future digital DRR education that will guide us to sustainable and resilient cities and communities.

## 2. Theoretical Background

### 2.1. The State of the Art of DRR Education in Japan

The ultimate goal of DRR education is behavioral change [16–18]. DRR education is quite well-developed in Japan and has its root in experiential learning from past disasters. The core of DRR education is the link between school, home and community [19–21]. School education alone cannot lead to DRR-related action, although it is important to provide basic knowledge and awareness. Real action starts when school, community, and family come together, and this is feasible when the students are in the 6 to 15 age group, which is the period of elementary and junior high school. A previous study [20] proposed the unique KIDA tree model of disaster education, which focuses on knowledge, interest, desire and action as a progressive process leading to risk reduction action. It also urged that some of the activities need to be conducted in school, some in the home and some in the community, and the connectivity of these three action areas is very important for successful DRR education.

In the case of Japan, there are different types of DRR-related education based on the needs and priorities of the local governments. There are four major milestones in the Japanese context of DRR education in recent years (Table 1). The first one started with the development of an early warning system after the Isewan typhoon in 1959. DRR 2.0 started after the 1995 Great Hanshin Awaji Earthquake, which highlighted the importance of self-help, mutual help and public help in DRR education. The experience of the 1995 earthquake has driven home the importance of the school–community linkages [6]. The next milestone for DRR education was related to DRR 3.0, which was the Great East Japan Earthquake and Tsunami of 2011. This was the largest earthquake on record and has shaken the whole DRR concept in Japan. It brought a concept of community-centric DRR education and the linking of different types of education, such as ESD (education for sustainable development), DRR education and environmental education, together under the common framework of school–community linkages [21,22]. We regard community-centric DRR as an important milestone for implementing SDG 11. "Network help" became a new concept added to the previous three types of help: self-, mutual and public. The concept of network

help is perceived within the community through neighborhood organizations and help provided by community volunteers, as well as linking the community with the external world [23], and the new perspective of DRR education was revisited to integrate network help with other types of help [22].

**Table 1.** Milestones of DRR education in Japan.

|  | Milestone | Components |
|---|---|---|
| DRR 1.0 | School–community linkage | Early warning, school as evacuation site |
| DRR 2.0 | Selfhelp, mutual help, public help | Enhance structural and functional capacity of school |
| DRR 3.0 | Community-centric and network help | Multi-hazard dimension of schools |
| DRR 4.0 | Responsible citizen behavior | Digital technology |

The new perspective of DRR education is linked to DRR 4.0, which connects to the digital era in Japan, known as Society 5.0 [6]. The current COVID-19 pandemic has added a new dimension to DRR education, which is safety education [11]. The pandemic has urged us to think about complex, multiple and cascading hazards and about its implications for DRR education. This was also exemplified in the recent IPCC report of 2022 [3], which emphasized the importance of climatic and non-climatic risk drivers together. School–community–family linkages, risk communication, and responsible citizen behavior are key variables to reduce the impacts of the pandemic and can be adapted to other disaster risk management needs [11]. The role of technology has become increasingly important in this COVID-19 scenario [24], and digital tools play a critical role not only for educational continuity but also in addressing DRR education.

*2.2. Digital Technology and SDGs*

Throughout the UN documents declaring the SDGs, the phrase "no one is left behind" occurs frequently [25]. The focus of the SDGs is different from that of the MDGs (Millennium Development Goals, the predecessor of the SDGs) in that SDGs are development goals that are essential for all humankind, while the MDGs focus primarily on developing country issues [26]. The SDGs have broader topics of interest, covering social, economic and environmental issues. Notably, the SDGs focus on the "*means and methods*" of achieving the given goals. We need to think how to implement these sustainability practices. Until recently, urban sustainability efforts focused on environmental sustainability. They targeted the use of renewable energy, resource reuse and technology to improve the efficiency of use, with an eye toward coexistence with the natural environment. Subsequently, the importance of not only the environment but also the continuity and sustainability of the business models (economies) of the businesses that provide services in cities, and the sustainability of society itself, specifically the communities and infrastructure created by people living in cities, have been recognized [27].

Although ICT use itself is not one of the 17 SDGs, it is mentioned that "The spread of information and communications technology and global interconnectedness has great potential to accelerate human progress" [25]. With regard to a link between digital technology and SDG 11, a frequently discussed topic is the smart city [28,29]. The International Organization for Standardization defines a smart city as a "*city that increases the pace at which it provides social, economic and environmental sustainability outcomes and responds to challenges such as climate change, rapid population growth, and political and economic instability by fundamentally improving how it engages society, applies collaborative leadership methods, works across disciplines and city systems, and uses data information and modern technologies to deliver better services and quality of life to those in the city*" [30]. It states that the use of information technology in solving city challenges will eventually guide us to being sustainable. The use of information/digital technology in disaster response has a large body of literature. A popular topic is management of disaster relief operations [31] and the use

of social media or other information delivery tools to help people in an affected area [32]. In DRR education, digital technology can be used to create new and innovative forms of narrative [33] and digitally enabled DRR education can respond to the specific needs of program participants [34].

Janowski [35] defines digital government evolution in four steps. The first step is digitization, where a governmental organization starts to utilize information technology within their institution, but transformation has not yet taken place. The second step is transformation, where internal government transformation—that is, process changes—occurs by utilizing those technologies. The third step is engagement, focusing on changes in the relationship between government and external stakeholders including citizens. The final step is contextualization, where transformation utilizing digital technology becomes context-specific. Disaster risk reduction and education are rarely discussed and we lack empirical data on how digital technology could enhance a disaster mitigation process [36].

## 2.3. The Role of Local Government

Japanese local governments are in charge of supporting residents to protect their lives in a disaster. They are responsible for creating a hazard map with information on vulnerability and supportive local community activities led by voluntary disaster prevention organizations, etc., on a daily basis. This approach differs from a seismic hazard map, which illustrates frequency, probability of occurrence, magnitude and intensity. Japanese local governments use a hazard map in the same way as the ISO definition of a hazard map. It is defined as "*a map developed to illuminate areas that are affected or vulnerable to a particular hazard (e.g., earthquakes, landslides, rockslides)*" [15]. A hazard map is an essential tool for DRR activities in everyday practice. According to a survey report by the Mobile Society Research Institute in Japan, the number of people who were aware of what a hazard map is was around 40 percent in 2021 (https://www.moba-ken.jp/project/disaster/disaster20210602.html, accessed on 23 May 2022). Moreover, the Ministry of Land, Infrastructure, Transport and Tourism of Japan reports that only 30 percent of people who acknowledge a hazard map recognize whether their house is in a disaster-prone area or not. As discussed previously, resident associations and voluntary disaster prevention organizations play essential roles in implementing self-, mutual and public help among citizens. A hazard map is a basis for increasing different types of help. Local governments help citizens to recognize and understand hazard maps of neighborhoods through daily local community activities.

Once a destructive phenomenon takes place, the most critical operation for local governments is to issue an evacuation order to citizens. They are also responsible for organizing disaster relief operations. Currently, evacuation orders from local governments are sent out by district. This makes it difficult for people to recognize the situation as their own. For instance, even when an evacuation order is issued by a certain city, required evacuation activities differ depending on the living location (Figure 1). Each household has different types of vulnerabilities to natural disaster. Family structure and other household circumstances affect people's actions. Some people may need assistance in evacuating, some may need medication for a chronic illness, some may live in an area identified as high risk on the hazard map (a warning area on the map), some may live near a designated evacuation center and some may need a certain amount of time to reach that evacuation center. DRR education is important to increase people's awareness about their own risks in a disaster situation.

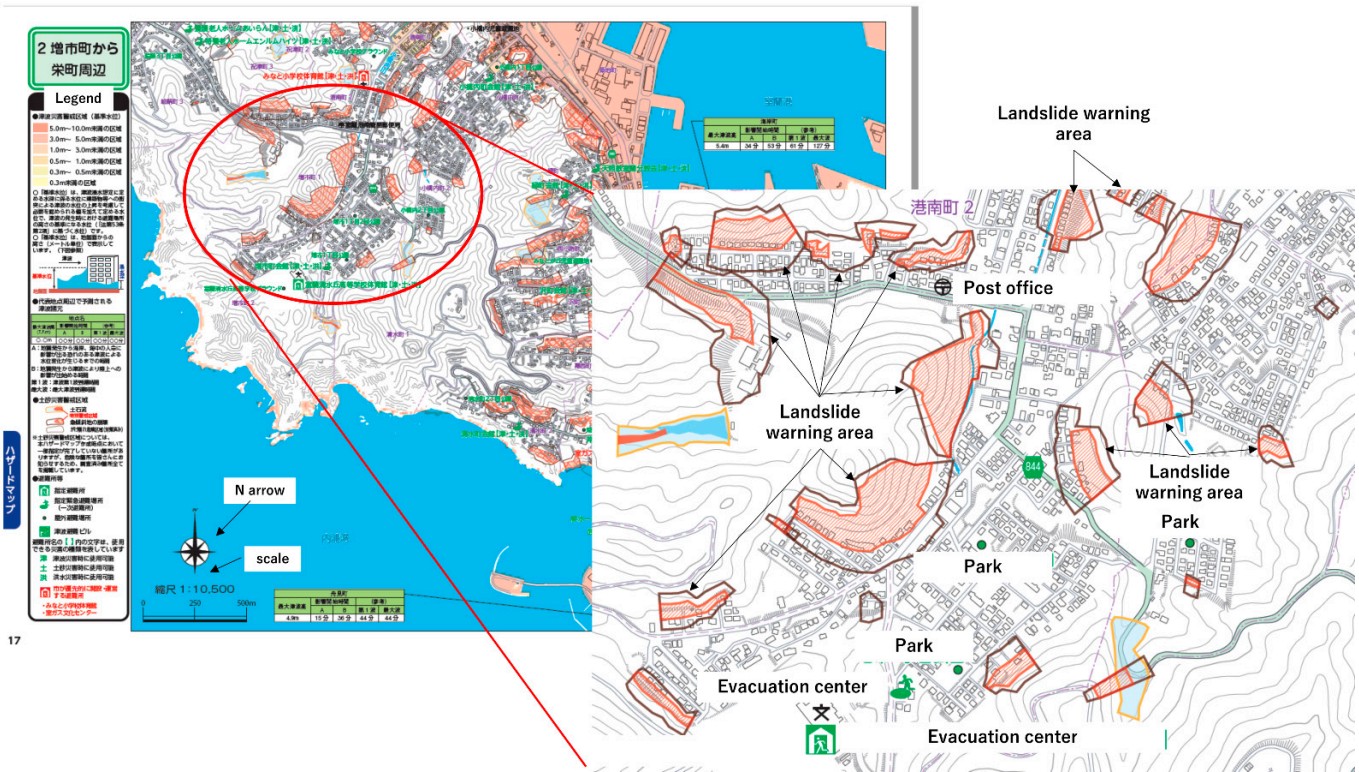

**Figure 1.** An official hazard map for a landslide in the *Masui* (district name) and *Sakae* districts of Muroran City (source: Muroran City; https://www.city.muroran.lg.jp/main/org3250/zentaizu.html, accessed on 23 May 2022).

This research shines a light on the role of local governments in enhancing community-centric DRR education and helps in understanding how digital technology works under a given context.

## 3. Methodology

### 3.1. Research Design

In this research we employed a design science methodology [37] and utilized action design research [38] of information systems. Design science is concerned with creating artifacts to achieve goals [39]. It aims to derive design requirements and systems features from practical observations. The design science approach begins with the notion of "the sciences of the artificial" advocated in [40]. The goals in design science research are determined based on problems. In this sense, the approach aims at problem solving and deals with generalized problems [41]. Design science incorporates the following steps: (1) awareness of a problem, (2) suggestion of a solution, (3) development of an artifact (information system) and (4) evaluation (feedback) [42]. We also take on the essence of an action design research approach, which focuses on more practical intervention. Action design research incorporates more practical relevance compared to the design science methodology [43]. This research follows the four steps of the design science methodology and includes practical intervention in the field of disaster education in a local community. For the evaluation step, we prepared two types of questionnaires. Noticing the importance of behavioral changes in DRR education, the first questionnaire investigated both mind and behavioral change. The second questionnaire investigated the degree of awareness about DRR. We used five categories to measure the level of DRR awareness: imagination, sense of crisis, mutual aid, interest and anxiety [44]. Like the KIDA (knowledge, interest, desire and action) model for DRR education [20], these five categories were developed based on the notions of knowledge and action, but with more focus on psychological preparedness for disaster.

Based on a design science methodology, we developed a series of workshops for DRR education employing digital technology. Workshops were hosted by a Japanese local government, Muroran City in Hokkaido. Muroran City was selected as one of the authors has been doing joint research in the university on disaster management and the use of digital technology for the last three years.

### 3.2. Research Context

Muroran City, in Hokkaido, Japan, has approximately 80,000 inhabitants. It is located in the southern part of Hokkaido, which is the largest island in the north of Japan (Figure 2). In September 2018, the Hokkaido Eastern Iburi Earthquake occurred in the southwest of Hokkaido. It measured 6.7 on the Richter scale, which is the largest ever in the Hokkaido area. The earthquake caused massive landslides and was followed by power and water outages. The number of deaths was 42 as of January 2019. In Muroran City, 2910 households were damaged due to power and water outages. Fortunately, Muroran City was spared from catastrophic damage, but it was an event that strongly reminded inhabitants of the need to be prepared for disasters. The results of a behavioral survey conducted by Muroran City after the earthquake showed that about 30 percent of the respondents (citizens) had not done anything at all to prepare for disasters, indicating the need to improve DRR education.

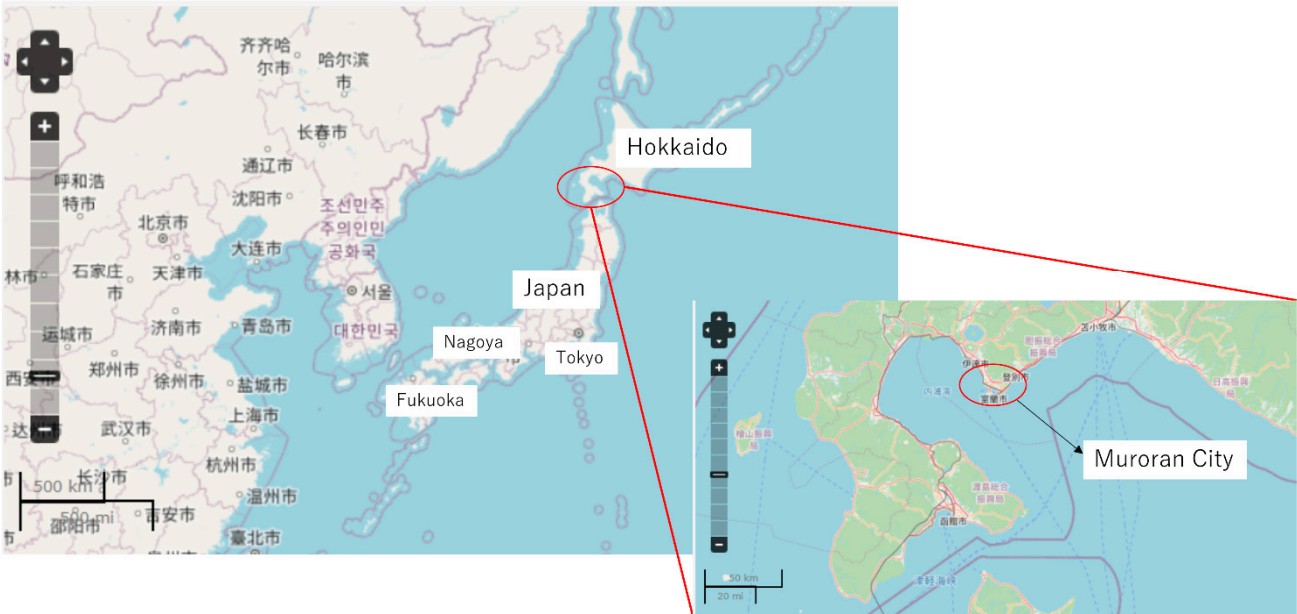

**Figure 2.** Location map of Muroran City (https://openstreetmap.jp/#zoom=4&lat=38.06539&lon=139.04297&layers=000B, accessed on 23 May 2022).

Traditionally DRR education in Muroran City has been provided through the "One-Day School for Disaster Prevention" program, led by the Hokkaido Board of Education, where local communities and schools learn about disaster prevention. Muroran City has been conducting disaster prevention drills to commemorate Disaster Prevention Day (1 September) and World Tsunami Day (5 November) and on 11 March, when the largest earthquake hit northeast Japan in 2011 (the Great East Japan Earthquake). In Muroran City, city officials support disaster prevention training and evacuation drills held by resident associations and voluntary disaster prevention organizations. Muroran City gives supplemental lectures on DRR and preparedness with the cooperation of local universities. In response to the behavioral survey after the Hokkaido Eastern Iburi Earthquake, city officials found it was necessary to train DRR leaders in each generation, in addition to supporting resident associations and voluntary disaster prevention organizations.

In 2017, the coverage rate of voluntary disaster prevention organizations in Muroran City was 61.3 percent. This was far below the national average, which was 82.7 percent. In order to improve the coverage rate, Muroran City decided to reform the voluntary disaster prevention organizations. Voluntary disaster prevention organizations are formed by resident associations, which are normally composed of people in their 60s and older. Originally, there were voluntary disaster prevention organizations in Muroran City; however, in order to make the most economies of scale, these groups were merged into 15 units. This enabled resident associations that were small or not very active in events to become members of voluntary disaster prevention organizations with the help of those that were active. At the same time, city officials were thinking about a generational approach. As discussed earlier, resident associations usually consist of the elderly, those over 60. These associations lacked DRR leaders and needed to strengthen ties to local communities. In order to move forward with this wide-area initiative, it was necessary to enhance DRR education in accordance with generational characteristics, including elementary school students, junior high school students, high school students, the working generation, and elderly people.

To begin with, Muroran City started to provide a disaster prevention day camp to elementary school students in 2019. A gymnasium of the Muroran nursing school was used for the day camp. The first floor of the gymnasium had been used as a stockpile for disasters. Students of the nursing school voluntarily participated in the program as part of a class on disaster medicine. Around 40 students from elementary schools in Muroran and neighboring municipalities have participated in the program every September since 2019. Muroran City envisions expanding this program to junior high school students. For those who are in the working generation (mainly 40–50s), a discussion on how to create a business continuity plan in a particular shopping district (named the *Nakajima* shopping district) started in 2021. In Muroran City, city officials noticed they had no link to high school students. Students in a high school become busier than elementary and junior high school students due to school sport/cultural club activities and cram schools for university entrance examinations. They hesitate to participate in extracurricular activities such as a local community DRR program. In addition, elementary and junior high schools have ties to local governments because they are compulsory education. However, high schools are not compulsory, so they are under the jurisdiction of a prefectural (regional) government. This makes official contact with local government (city officials) difficult. In many cases, governments rely on each city official's personal connections with high school teachers.

*3.3. Instruments Developed*

While considering the approach to DRR education for different generations, Muroran City participated in joint research that is being led by one of the authors. The joint research consists of multiple local municipalities and private companies. The consortium has discussed the use of information and communication technologies (ICT) in disaster-related local government activities. Disaster education is one of the featured topics. Consortium members discussed how to benefit from ICT and digital technology use in disaster education. A city official of Muroran was interested in a presentation made by Salesforce Japan, which was also a member of that consortium. The presentation introduced "myTrailhead", a cloud-based online education platform. The city official found this platform could be useful for a disaster education program for high school students, whom the city had tried to approach. The official said; *"Since high school students are familiar with various information technologies on a daily basis, I believe that the platform could be used in a disaster education program for them".*

A three-day series of DRR education workshops were developed for students in the *Muroran-Higashi* high school. The workshops started in November 2021 and a one-day workshop was conducted every three weeks until December. The program introduced an active learning methodology that made students more active in the learning process. Active learning methods are becoming popular in Japanese universities. They require "students to take an active part in their leaning, rather than being passive while the teacher lectures or

engages in other forms of direct teaching" [45]. A total of 28 students from the first and second years of the *Muroran-Higashi* high school participated in the workshop.

"myTrailhead" is an online education platform developed by Salesforce. This platform consists of a set of modules that function as a basis for independent study. Each module contains texts, supplemental materials (image, video and other graphic items) and a set of quizzes for reflection. Users are supposed to finish each module in 10 to 15 min. Once they clear a quiz section, it is possible to proceed to the next module. "myTrailhead" introduces gamification, and users can learn at their own pace. At the workshop in Muroran, students were divided into six groups and tried to create their own module for DRR education. Active learning requires students not only to study the educational contents provided (passive learning) but to learn what they want to know and create contents from scratch. Each group decided on a content theme for their module and collected the necessary materials from the Internet. There were lectures from a disaster prevention specialist, certified by the Japan Disaster Reduction Professionals Organization, and an academic researcher on the first day (Figure 3). Instruction of how to use "myTrailhead" followed afterwards. On the second day, students formed groups and started to create their own content for DRR (Figure 4). On the third day, which was the final day of the workshop, students continued the content creation and developed a presentation (Figure 5). Students presented their content in three minutes, and they were evaluated by the mayor and other lecturers.



**Figure 3.** Structure of the workshop.

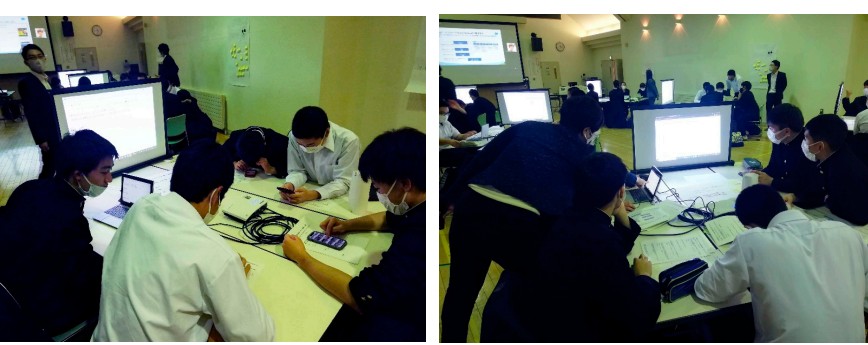

**Figure 4.** Discussions in each student group.

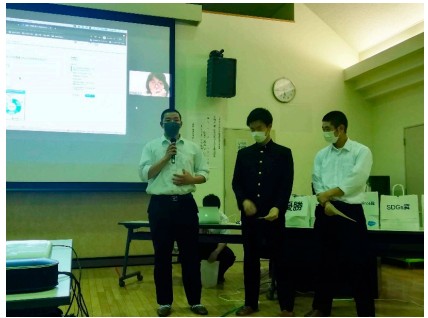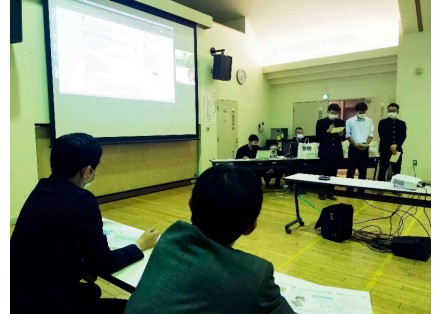

**Figure 5.** Student presentation and evaluation.

The winning team created content focusing on disasters caused by heavy rain (Figure 6). They set the question, "How much rain is 50 mm?" Students started with a definition of what 50 mm per hour means in a disaster context. They found that with 50 mm of rain per hour, umbrellas would be useless. Landslide disasters can be caused by such rain. They said, "*If you find a cliff face, hillside, or mudbank, there is a high possibility that a landslide will occur there. If you pay attention to these things, you will be less likely to encounter a disaster*". They also showed that the number of disasters caused by rain is increasing. One of the group members stressed the importance of everyone being aware of global warming, which is a cause of heavy rainfall. At the end of their presentation, they introduced an application that uses AR (augmented reality) to visualize the level of rainfall and said, "*Rain gives us blessings, but sometimes it also endangers our lives. Let's be prepared for rain-related disasters*".

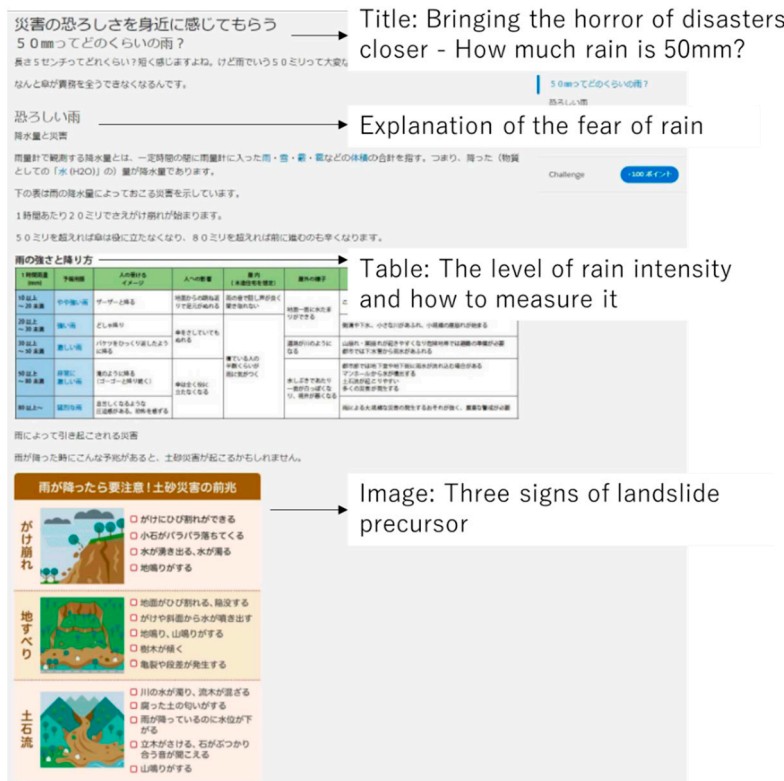

**Figure 6.** Content created by the winning team focused on disasters caused by rain.

Another group, which received a mayor's award, focused on disaster preparation for women (Figure 7). They said, "*Are you sure you have everything which women would need for a long-term evacuation?*" They started their presentation with a call to action. The presentation stressed emergency items especially for women in an evacuation center. The group created a list of things to prepare for women who are pregnant and with young children and said, "*Gentlemen, please listen so that you can take care of your mothers and future wives!*" They also found out how women spend time in an evacuation center. They urged the audience to review their emergency bags and prepare the necessary items that only women can take in their daily shopping.

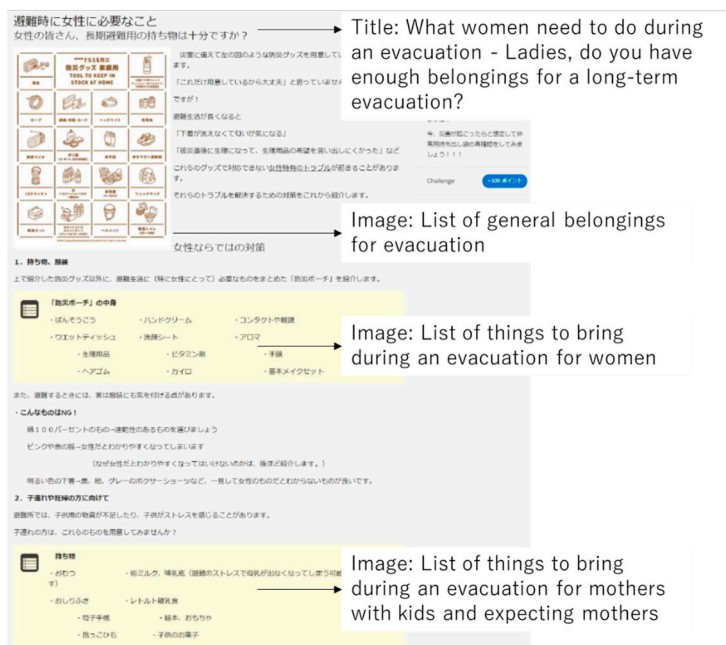

**Figure 7.** Content created by the mayor's award team focused on emergency items for women.

## 4. Analysis of the Results

We used the following five conceptual categories advocated by [44] to analyze how the workshop changed participants' minds and behaviors. These categories show the degree of awareness of DRR.

(1) Imagination of the disaster situation—the ability to imagine what would happen, what would be needed and what would be done if a disaster occurred. If you have a low level of this skill, learn about disasters and develop your imagination by studying disasters and listening to the stories of disaster victims.

(2) Sense of crisis regarding disasters—this indicates how seriously you take disasters and whether you think the current situation is bad. If your score is low, you should be aware of the fact that disasters can happen tomorrow and acquire a sense of crisis.

(3) Awareness of mutual aid—this describes the mind that wants to do something for society and others. Disasters cannot be overcome without the cooperation of everyone in the community. If you have a low level of this, you should reaffirm the importance of mutual aid and think about others.

(4) Interest in disasters—the degree to which a person is interested in disasters and sees them as a personal matter. If interest is low, you are indifferent to disasters. Try to consider disasters as your own problem and think about what you can do to prevent disasters.

(5) Anxiety—the degree to which you are worried about disasters. Anxiety can be a driving force for disaster preparedness, but unlike categories 1 to 4, it should not be too high.

Two types of questionnaires were prepared: (1) open questionnaire asking about mind and behavioral changes and (2) a rating scale asking about the degree of the above five categories. First, we present the results from the open questionnaire. We conducted this questionnaire after a set of three workshops. Sentences underlined by the authors are closely related to each category's definition.

Imagination of the disaster situation:

*"It was really new for us to create our own content and present it to other people. I thought it was a very good idea to not only listen to a talk about disaster prevention, but also to create our own content and communicate it to others in the group. In our group, we*

*studied precipitation. We learned a lot about what kind of disasters tend to happen with precipitation of 50 mm per hour, and what kind of disasters tend to happen with this level of precipitation. I think we will encounter a lot of rain-related disasters in the future, and it is good to learn more about how to prepare for them."*

*"It was simply fun. I learned a lot of new things, such as how to make the content easier to understand by using bold letters, and how to make it easier to read by having 1–2 lines of text between the lines. My group looked at measures for women. I had never thought about the measures needed for women before, so it was a new lesson for me. Listening to the other teams' presentations, I learned a lot about disaster prevention from a different perspective, such as floods and dangerous places to evacuate."*

*"I used to be unable to think of disasters as something close to myself, but as I did my own research and looked at various data, I gradually became more familiar with them."*

*"I would like to prepare an emergency supply bag."*

*"I will check hazard maps."*

Sense of crisis regarding disasters:

*"By learning on my own, I was able to better understand the probability of earthquakes, which strengthened my belief that we never know when they will occur."*

*"I had thought about countermeasures only for earthquakes, but not for floods, land-slides, tsunamis, etc. I now feel that I should think about disaster prevention from various perspectives."*

*"By proactively researching disasters, my awareness of disaster prevention has increased dramatically."*

Awareness of mutual aid:

*"The simple problem of reducing or eliminating the number of victims of disasters is difficult to solve, and I learned the importance of not only self-help but also mutual aid, which gave me the idea to value cooperation with others."*

*"I became more conscious of not only thinking about disaster prevention on my own but also communicating it to others."*

*"I will anticipate possible disasters and promote information sharing with family members in case of emergency."*

Interest in disasters:

*"I will try to understand specific information about disasters that have happened so far."*

*"I will identify places near my house where secondary disasters are likely to occur."*

*"I will confirm location of evacuation sites, share information with family members, and prepare disaster supplies. Preparing my mind for the possibility of a disaster right now or tomorrow."*

*"I started to look for dangerous places on my way home."*

We did not find any comments related to anxiety from the open questionnaire. For the quantitative evaluation of the workshop, we handed out the second questionnaire before and after the workshop. This enabled us to analyze changes in participants' awareness. The second questionnaire employed 20 questions, with sets of four questions corresponding to the five categories of DRR consciousness. Participants responded to each question on a six-point scale: "6" meant that it applied very well while "1" meant that it did not apply at all. A set of questions were provided by the National Research Institute for Earth Science and Disaster Resilience Japan [44]. Each of the five categories was worth 24 points. There were 120 points in total. The overall results for the second questionnaire and paired *t*-test results are shown in Figure 8 and Table 2. Notably, we found significant changes, especially

in the imagination category with a five-point increase. Scores for other categories also increased after the workshop. The total score increased by 11 points.

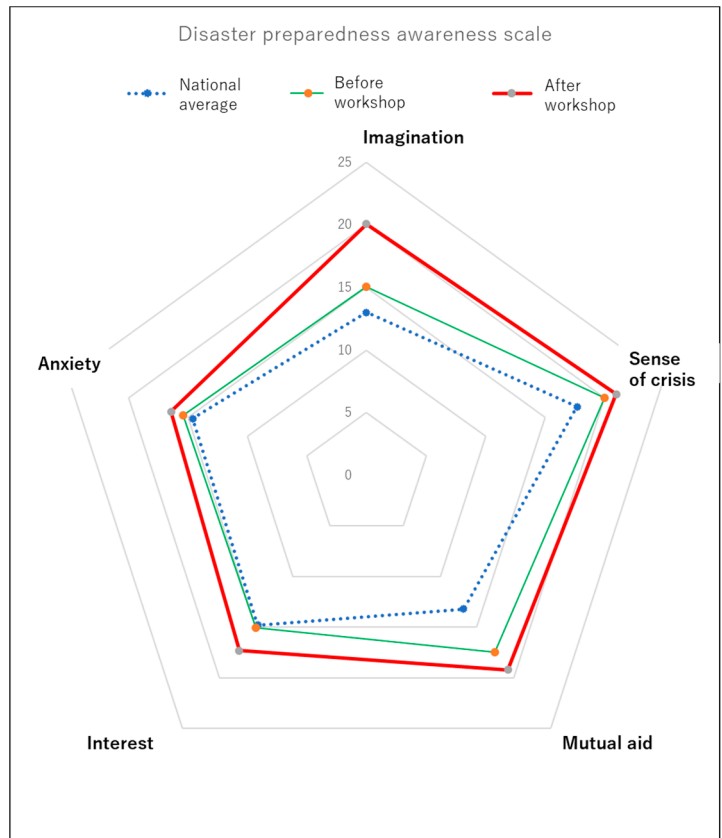

**Figure 8.** Results of disaster-preparedness awareness survey.

**Table 2.** Paired *t*-test results.

|  | Before Workshop | | After Workshop | | Mean Differences | *t*-Value | |
|---|---|---|---|---|---|---|---|
|  | Mean | S.D. | Mean | S.D. | | | |
| Total | 82.62 | 10.69 | 94 | 10.09 | −11.38 | 6.12 | ** |
| Imagination | 15.15 | 2.46 | 20.08 | 2.16 | −4.92 | 7.24 | ** |
| Sense of crisis | 19.95 | 2.73 | 20.96 | 2.39 | −1.12 | 1.52 | |
| Mutual aid | 17.54 | 3.96 | 19.23 | 3.93 | −1.69 | 3.91 | ** |
| Interest | 14.96 | 3.18 | 17.31 | 2.8 | −2.35 | 4.53 | ** |
| Anxiety | 15.12 | 3.9 | 16.42 | 3.92 | −1.31 | 1.74 | |

** $p < 0.01$.

We conducted a paired *t*-test for the questionnaire results before and after the workshop Highly significant differences in total score ($p = 0.000$), imagination score ($p = 0.000$), mutual aid score ($p = 0.000$) and interest score ($p = 0.000$) before and after the workshop were confirmed. Differences for the sense of crisis score ($p = 0.141$) and anxiety score ($p = 0.094$) were non-significant. We confirmed a highly significant difference in the total score, and thus a set of three workshops increased participants' overall disaster-preparedness awareness. In particular, awareness regarding imagination, mutual aid and interest was changed by workshop activities. These results were supported by the first questionnaire, where we obtained detailed descriptions of how participants felt their mind and behavior changed after the workshop.

## 5. Discussion

### 5.1. Practical and Theoretical Implications

The case of the DRR workshop in Muroran City was unique in that it employed digital technology to involve high school students in a disaster education program. The practical implications of this research are twofold. Firstly, the framework of the workshops worked well for high school students, who are in a black box of disaster education. Secondly, the results of the questionnaire survey showed that participants' minds and behaviors towards DRR changed after the workshop, which is one of the ultimate goals of DRR education. Participants gained the ability to imagine what they need to do in the case of a disaster, the ability to acknowledge the importance of mutual aid and the ability to consider disasters as a personal affair. What is currently being argued in Japan is that people do not act to save their own lives, even when local governments issue an evacuation order. Consequently, we lose lives that could have been saved if appropriate evacuation actions had been taken. While recognizing the importance of awareness of hazard maps, we argue that how people understand or interpret the information given by the map is equally important. It is said that decision making in a disaster situation is prone to normalcy bias, in which people believe the situation is not serious enough and they will be safe [46,47]. The above three abilities are important to overcome normalcy bias.

This argument aligns well with digital transformation discussions, such as in [48–50]. The theoretical implication of this research is that the project demonstrates a potential approach for DRR 4.0 involving the use of digital technology. A conventional goal in IT-enabled organizational transformation is optimizing operational processes and achieving efficiency, but this will change in the era of digital transformation [48]. In the context of digital transformation, consumers and citizens use digital technology in order to transform themselves [49]. Digital infrastructures can promote individuals' capacities and enhance their autonomy [50], which results in people doing things by themselves. These discussions align with what we find in the workshop. As discussed in [33–35], digital technology enables DRR education to be more context-specific and thus supports people in improving their disaster-preparation awareness. As we see in Figures 6 and 7, a digitally based DRR education program personalized disaster-related knowledge and succeeded in the creation of context-specific educational materials. The content creation process enabled participants to reorganize their minds and behaviors. Through the workshop, participants incorporated what they learned into practical actions for future preparedness.

We believe contextualization would be essential to achieve the SDGs' aspiration that "no one is left behind". This is both a practical and theoretical implication of this research. As discussed previously, the scope of evacuation orders from local governments are district-based. Each district faces different disaster risks and household/personal circumstances. Conventional, paper-based information provision is too labor-intensive to be tailored to each citizen's circumstances. By creating residents' own (context-specific) DRR content through the digitally based DRR educational platform, each citizen could evaluate and understand their own risks. Subsequently, this could contribute to raising people's awareness and modifying their behavior. It could even change how people interpret warnings or evacuation orders issued by local governments. These arguments encourage us to regard digital technology as an essential means for the implementation of SDGs. A cloud-based education platform should be introduced not to optimize organizational processes but to generate a collaboration space for the local community (high school students in this case) and to guide a co-creation process for DRR education. The city of Muroran found a new way to engage with high school students. In the near future, they plan to bridge the inter-generational gap by utilizing what high school students have created. The city will set up a council to discuss how to increase the number of members in resident associations.

*5.2. Limitation and Future Research Directions*

This research has a couple of limitations. The workshop was held in only one city and we discuss only a single case. Even though Muroran City's case shows that the workshop works for changing participants' minds and future behaviors, we need another city where this workshop method can be applied. Fortunately, one of the city officials who is a member of the research consortium plans to introduce the same method into another city's DRR education program in 2022. This city envisions spreading DRR education to people after compulsory education, especially the working generation. The city official believes digital technology could help deepen DRR education and change people's mindsets. Another aspect we need to consider in future research is the comparison between digitally based DRR education and non-digital DRR education programs. We believe further investigation into how digital technology could enhance peoples' willingness and ability to take life-saving actions will help our understanding and provide more data on how we can realize DRR 4.0.

## 6. Conclusions

In order to make cities and human settlements inclusive, safe, resilient and sustainable, which is advocated by SDG 11, better adaptability and preparedness among the local community are the most important factors. Technology should be utilized to strengthen local knowledge and familiarity with local conditions [24]. One area where digitalization can be expected to be contextualized in line with the SDGs is disaster preparedness and response [51]. Looking at the Japanese context, the national government released a vision called "Society 5.0" as part of the fifth Science and Technology Basic Plan 2016. This vision targets society after 2030, stating that, "New value creation through innovation will eliminate disparities based on region, age, gender, and language. It enables a detailed response to the diverse and latent needs of individuals" (https://www8.cao.go.jp/cstp/english/society5_0/index.html, accessed on 23 May 2022). The vision aligns well with the philosophy of the SDGs and therefore we need to understand how digital government-related activity could enhance people's quality of life. As this paper shows, local government is in charge of arranging voluntary disaster prevention organizations and helping residents to be well-prepared for future disaster situations. In the last two years, through 2020 and 2021, there have been several contemporaneous disasters, and this has brought a new dimension involving multi-hazard risk scenarios. The traditional approach to earthquake-, flood- or typhoon-related education is possibly not effective anymore. We need to think of this complex risk landscape, and approach it collectively with different types of risk knowledge and actions.

Local authorities must involve every citizen in DRR education actions. Digital technology can help local governments reach people who are not usually part of local community activities. This is a new process of creating value by digital technology [52], which we call "orchestration" among the diverse players [53] of disaster management. Digitally based DRR education programs enable context-specified DRR education, changing people's awareness and behavior. Along with the vision of the Japanese "Society 5.0", we believe digitally enabled DRR education could be the first step for future personalized digital government services [54,55]. Once such digital personalized services are available, people in disaster-prone areas would be able to receive tailored warnings and behavioral direction for themselves. We believe this could guide us to an inclusive, safe, resilient and sustainable community and city, where people's ability to cope with the unexpected evolves over generations.

**Author Contributions:** Conceptualization, M.S. and R.S.; Methodology, M.S.; formal analysis, M.S.; writing—original draft preparation M.S. and R.S.; editing, M.S. and R.S. All authors have read and agreed to the published version of the manuscript.

**Funding:** JSPS KAKENHI Grant Number JP 21K18019 to M.S.

**Institutional Review Board Statement:** Not applicable.

**Informed Consent Statement:** Informed consent was obtained from all subjects involved in the study.

**Data Availability Statement:** Not applicable.

**Acknowledgments:** The authors acknowledge the dedicated support provided by Muroran City and Salesforce Japan to this study.

**Conflicts of Interest:** The authors declare no conflict of interest.

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
