# Peer review of "The Potential of Digitally Enabled Disaster Education for Sustainable Development Goals"

_sustainability, doi:10.3390/su14116568_

Round 1

Reviewer 1 Report

I recommend rejecting this article, based on multiple motives:

  • The authors misuse terms like risk, disaster, hazard or vulnerability, making it clear they do not fully understand their meaning.
  • The paper lacks scientific consistency. It is merely a description of a locally applied Disaster Risk Reduction project.
  • The paper lacks proper structure: there are no methodology (properly done) and results sections. I understand that it is not a classical research article, but I think its submission category needs to be changed.
  • The cartographic material is very poor and should be greatly improved. So is the commentary regarding the “hazard map” (Figure 1). I included some comments that should help the authors to clarify what hazard maps and vulnerability maps are.

I included some comments that should help the authors improve their work, referring to terminology issues, the misuse of English language and confusing phrasing. Furthermore, I strongly recommend they rewrite the paper and submit it again to a Q3/Q4 journal.

Author Response

Thank you very much for useful and helpful comments and suggestions. Please find our responses as enclosed. Please note that each line number in our responses corresponds to the revision file that is seen as a track change mode.

Reviewer 2 Report

Thank you for the opportunity to review the paper. It is an interesting piece of work dealing with an important issue. The paper is already quite well prepared therefore I would suggest publishing it after some revisions.

Below you may find some more or less specific comments.

Line 48-50: “The youth of today are leaders of tomor-48 row, the so-called “Generation Z”, who were born between mid-1990s and 2010 are con-49 sidered to be digital natives.” Please give a reference here.

Line 53: “(…) the governance mechanism to support the technology 53 is still going on at a slower pace.” – on what basis do you state that? Where this conclusion comes from? Although it is generally right what you say here, you should mention that at least on the discursive level the governance mechanisms in local communities are to support innovations and new technologies while dealing with natural hazards (ie. doi.org/10.3390/su14042052 ).

Lines 69-77: This part is really interesting. This is describing an event of more then 20 years ago. I am curious whether this issue changed in years. For example how about the Great East Japan Earthquake off 2011? I suppose there were a lot of researches on that huge event. And also, how about smaller events? Like local flooding, extreme meteorological events? Does it differ? Maybe a comment on that issue here or in the discussion part would enrich the paper.

Line 240: “Muroran City, city officials noticed they had no link to high school 240 students.” – Why is that?

Line 424: missing reference

Author Response

(The authors gave the same response as above.)

Reviewer 3 Report

The manuscript is very interesting and has a potential to be published. Suggestions for the manuscript improvements are:

  • The Abstract and Introduction sections should provide the main research goal. 
  • The structure of the manuscript should be improved. Section 2 should be Theoretical background. Section 3 - Methodology, 3.1 Research design - provide research questions or hypothesis, 3.2 Research context, 3.3 Instruments, 3.4 Sample descriptions, Section 4- Analysis of the results. The results should be better presented and analyzed using adequate statistics. Section 5 - Discussion and Conclusion. In this section please provide implications and the limitations of the research, and scientific contribution. Explain why this research is important and is there possibility to use this approach in other cases. 
  • List of References should be extended with newer references from the last three years. Some of the references are not well inserted in the text (page 11, section 4.1 has in the text (Error! Reference source not found.)
  • English language should be improved. 

Author Response

(The authors gave the same response as above.)

Reviewer 4 Report

Hello,
I had a good time reading your research paper. The idea is interesting and would encourage this to be bought to light. However, this paper would need more editing and reviewing. Do not lose hope. You are almost there! 

One missing is on identifying the previous work having been done. Please refer to here https://scholar.google.co.id/scholar?hl=en&as_sdt=0%2C5&authuser=1&q=Digitally+Enabled+Disaster+Education+&btnG=. Then some relevant previous works could be added and combined in the literature section. Definition of terms is also missing. Please add all definition on the literature section as well. All the best.

Author Response

(The authors gave the same response as above.)

Round 2

Reviewer 1 Report

I appreciate the efforts to restructure and improve the manuscript. However, it is still not suitable for publication in a journal like Sustainability. I have to reject it from publication again, due to reasons explained below. I recommend the authors to further improve it and send it to a Q3/Q4 journal, because it does not meet the standards of Q1/Q2 journals.

  • As I highlighted before, the authors fail to explain basic concepts of hazard, risk, vulnerability, which makes me doubt their understanding of it. This is one of the main reasons to reject again the paper. I do not wish to criticise in a mean manner, but the manuscript did not convince me that its writers understand the very notions they are writing about.
  • The article needs extensive English editing, as there are many grammar and stile mistakes. The manuscript is very hard to read, which greatly impacts on its evaluation.
  • Table 1 is not necessary, as it contains very little information. A diagram would be more on point and add esthetic value to the manuscript.
  • Figure 1 was not properly modified, and it still infringes on basic cartographic rules. This is not a hazard map, as I previously mentioned. Hazard maps express particularities of hazards (frequency, intensity, magnitude etc.). Also, it is not a vulnerability map. It is merely a location map of the study area.
  • Section 3.2. (which deals with the study area) lacks a location map of the study area.

Author Response

Sustainability_1681282_R2 Authors’ response to the reviewers

Thank you very much for useful and helpful comments and suggestions. Please find our responses as follows. Please note that each line number in our responses corresponds to the revision file that is seen as a track change mode.

Comments from Reviewer 1

1) As I highlighted before, the authors fail to explain basic concepts of hazard, risk, vulnerability, which makes me doubt their understanding of it. This is one of the main reasons to reject again the paper. I do not wish to criticise in a mean manner, but the manuscript did not convince me that its writers understand the very notions they are writing about.

(Response)

We rely on definitions of hazard, disaster risk reduction and vulnerability which are advocated by UNDRR. Given definitions are italicized in the text.

2) The article needs extensive English editing, as there are many grammar and stile mistakes. The manuscript is very hard to read, which greatly impacts on its evaluation.

(Response)

We have done another round of proofreading by a native English speaker. In order to improve readability, we broke up the longer sentences with over forty words into shorter ones. In addition, we adjusted some punctuation marks to change the rhythm of the sentences, and used some synonyms for words that are used several times in the text.

3) Table 1 is not necessary, as it contains very little information. A diagram would be more on point and add esthetic value to the manuscript.

(Response)

We deleted Table 1.

4) Figure 1 was not properly modified, and it still infringes on basic cartographic rules. This is not a hazard map, as I previously mentioned. Hazard maps express particularities of hazards (frequency, intensity, magnitude etc.). Also, it is not a vulnerability map. It is merely a location map of the study area.

(Response)

Figure 1 is derived from the official hazard map of Muroran City. City officials and citizens use this map for everyday DRR activities. In the revised manuscript, we explained how Japanese municipalities use a term hazard map. They follow the definition of ISO. We cited the document as reference #15. We added caption to Figure 1. That caption makes it clear that this is an official map created by Muroran City.

5) Section 3.2. (which deals with the study area) lacks a location map of the study area.

(Response)

We created Figure 2 under section 3.2.

Round 3

Reviewer 1 Report

I congratulate the authors on implementing the suggested modifications. Starting with the first reviewed variant, they changes the stucture of the paper and greatly improved it. There are only a few changes they should make in order to get the accept for publication, as stated in the comments inserted in the attached pdf.

I should mention that I truly appreciate their work and efforts, as well as their perseverance. I did not wish to give them a hard time, but only to help them make the most of this research work.

Author Response

Sustainability_1681282_R3 Authors’ response to the reviewers

Thank you very much for useful and helpful comments and suggestions. Please find our responses as follows. Please note that each line number in our responses corresponds to the revision file that is seen as a track change mode.

Comments from Reviewer 1

1) I advise the authors to reframe the results part (lines 18-19) to make the paper more appealing to be read. Also, this phrase may be subject to questions on its validity and accuracy.

(Response)

We rephrased the sentence as follows.

Three out of the five categories of DRR consciousness increased after the workshop, namely, the degree of imagination, mutual aid, and interest. We observed that participants’ mindset and behavior changed during the workshop activities.

2) These references should include a year. I am familiar with them and I know for a fact that UNDRR changed these definitions in time, so it is important to specify the time-frame the authors refer to.

(Response)

We changed the sentence as follows.

In 2017, the United Nations Office for Disaster Risk Reduction (UNDRR) defined disaster risk reduction as “preventing new and reducing existing disaster risk and managing residual risk, all of which contribute to strengthening resilience and therefore to the achievement of sustainable development [10].”

3) The hazard maps do not include vulnerability elements. Please delete the vulnerability part in order to ensure accuracy.

(Response)

We deleted accordingly.

4) I appreciate that the authors made clear what the ISO definition of the hazard map is. However, in order to provide accurate information, they should also mention that this approach is not common. Hazard maps illustrate hazard parameters like frequency, probability of occurrence, magnitude, intensity etc. The ISO definition relies on vulnerability elements, which are something else, as the definition provided by UNDRR states. Please include this specifics in the text.

(Response)

We added the following sentence in the beginning of section 2.3.

This approach differs from a seismic hazard map, which illustrates frequency, probability of occurrence, magnitude and intensity.

5) There is no year for this reference.

(Response)

We amended reference 10.